

# The importance of nitrogen fixation to a temperate, intertidal embayment determined using a stable isotope mass balance approach

Douglas G. Russell[1], Adam J. Kessler[1], Wei Wen Wong[1], and Perran L. M. Cook[1]

[1]Water Studies Centre, Monash University, Clayton, 3800, Australia.

*Correspondence to: Douglas G. Russell (douglas.russell@monash.edu)*

**Abstract.** The balance between denitrification and nitrogen fixation is the key control of the availability of nitrogen in coastal ecosystems and thus the primary productivity of these environments. However, evaluating the importance of denitrification and nitrogen fixation over large spatial and temporal scales is problematic. In this study, a combined mass and

stable isotope balance of nitrogen was used to constrain the cycling of nitrogen in Western Port, Victoria – a temperate, intertidal embayment in south-eastern Australia. This method is a more effective approach compared to the extrapolation of discrete measurements and geochemical approaches. The validity of the isotope and mass balance model has been tested by comparing the output of the model with the average measured isotopic signature of the sediment in Western Port. Using previously measured rates of nitrogen fixation and denitrification in combination with the isotopic signature of nitrogen

inputs from the catchment, atmosphere and the marine environment, the model returned an isotopic signature of $4.1 \pm 2.5$ ‰. This compares favorably with the average measured isotopic signature of the sediment of $3.9 \pm 1.2$ ‰. Sensitivity analysis confirmed that it was the isotopic values of the end-members, fractionation factors of assimilation and denitrification that exerted the greatest control over the isotopic signature of the sediment and not the loadings of the source and sink terms. Analysis of the relative importance of the various nitrogen inputs into the bay suggests that nitrogen fixation contributes 36

% of the total nitrogen inputs to Western Port.

## 1 Introduction

The availability of nutrients such as nitrogen plays a crucial role in regulating the primary productivity of many marine environments, particularly those in temperate regions (Herbert, 1999). In environments where the bioavailability of nitrogen is low, the limitation of autotrophic growth has been commonly reported (Elser et al., 2007). These effects can also be

transferred to higher trophic levels as a result of their place at the bottom of the food web (Field et al., 1998). Eutrophic conditions on the other hand are characterized by high nutrient concentrations and often results in the formation of widespread algal blooms. This not only leads to shifts in the community compositions (McGlathery et al., 2007), but can result in widespread hypoxia and loss of biodiversity resulting from the reduction in dissolved oxygen concentrations (Vaquer-Sunyer and Duarte, 2008).




Biogeochemical processes such as nitrogen fixation and denitrification are key mechanisms that help to regulate the bioavailability of nitrogen in marine environments. Nitrogen fixation results in an increase of bioavailable nitrogen in these environments, with organisms such as cyanobacteria and sulfate-reducing bacteria able to convert atmospheric nitrogen ($N_2$) into ammonium ($NH_4^+$) (Herbert, 1999). In contrast, denitrification is a removal process, whereby bioavailable nitrogen

($NH_4^+/NO_X$) is converted back into $N_2$ (Herbert, 1999). As these processes represent competing production and removal pathways the imbalance of either process can potentially lead to increased availability or a depletion of bioavailable nitrogen (Capone and Knapp, 2007), and are a key determinant whether marine ecosystems become oligotrophic or eutrophic.

On a global scale, the removal of bioavailable nitrogen by denitrification has been found to exceed the production through

biologically-mediated nitrogen fixation in marine environments (Gruber and Galloway, 2008). However, the degree to which denitrification dominates is a matter of much debate. Estimates at the low end suggest that rates of denitrification are 32–36 % higher than nitrogen fixation (Gruber, 2004; Gruber and Sarmiento, 1997), whilst Codispoti (2007), Codispoti et al. (2001) and Codispoti and Christensen (1985) found that denitrification is occurring 3 to 5 times faster than nitrogen fixation. On a local scale, the relative importance of denitrification and nitrogen fixation has been found to be more variable. Studies

by Cook et al. (2004), Risgaard-Petersen et al. (1998) and Welsh et al. (2000) found that nitrogen fixation dominates denitrification in some vegetated and non-vegetated temperate environments. Whereas, in a range of sub-tropical environments, Eyre et al. (2011) found that denitrification was invariably favoured over nitrogen fixation. Similar findings were also reported by Seitzinger (1988) who found that for a range of environments, including coastal marine settings, denitrification and not nitrogen fixation was the dominant process.

Clearly, site specific differences are the likely causes for the differing findings regarding the relative importance of these competing processes in marine environments. This is particularly the case with respect to the availability of nitrogen, with increasing concentrations of bioavailable nitrogen having been shown to have an inhibitory effect on the rate of nitrogen fixation (Howarth and Marino, 2006; Welsh et al., 1997). Similarly, it has been reported that there is a positive correlation

between the rate of denitrification and the concentration of bioavailable nitrogen (Herbert, 1999). The loading of organic material has also been shown to have a profound effect on the relative importance of denitrification and nitrogen fixation. Studies by Fulweiler et al. (2007) and Fulweiler et al. (2013) showed that after the direction of the net flux of $N_2$ could be reversed from net nitrogen fixation to net denitrification through an increase in organic material deposition.

Methodological issues surrounding the measurement of these processes can also account for the variability in the relative contribution of denitrification and nitrogen fixation in marine environments (Brandes and Devol, 2002). With respect to nitrogen fixation, commonly the acetylene reduction assay and labelled $^{15}N_2$ have been used to make discrete measurements (Mahaffey et al., 2005). However, these two methods give differing results: the acetylene reduction assay estimates gross nitrogen fixation, whereas the label $^{15}N_2$ approach estimates net nitrogen fixation (Gallon et al., 2002; Mulholland, 2007).



Geochemical inferences of nitrogen fixation have also found widespread use, with the assumption being made that any deviation from the Redfield N:P ratio of 16:1 (Redfield, 1958) is a result of either of denitrification and/or nitrogen fixation (Michaels et al., 2001). This approach is problematic, as there are processes not associated with denitrification or nitrogen fixation that can alter N:P ratios (Karl et al., 2002), such as the release of particulate-bound phosphorous under changing

redox conditions. Therefore, if the rates of denitrification and nitrogen fixation are comparable, then the nitrogen anomaly and hence inferred nitrogen fixation or denitrification will be undetected (Karl et al., 2002).

In addition to the methodological issues surrounding the use of geochemical inferences to measure denitrification in marine environments, there exist other methodological issues surrounding the use of the acetylene block technique. As this

technique has previously been found to under-estimate the contribution of coupled nitrification-denitrification to the total rate of denitrification, owing to the acetylene-induced inhibition of nitrifying microbes (Knowles, 1990; Koike and Sørensen, 1988). Whilst, more up to date techniques such as the $N_2$:Ar method and isotope pairing technique don't suffer the same drawbacks as the acetylene block technique, they still can return different estimates for the rate of denitrification. The $N_2$:Ar technique measures the net amount of denitrification, whilst the isotope pairing technique estimates the gross amount of

denitrification (Eyre et al., 2002).

The extrapolation of nitrogen fixation and denitrification rates from discrete measurements to larger spatial scales, even in the case of local studies can also account for the variability in the relative importance of these processes considered in previous studies. For instance, Mulholland (2007) found that a wide range of global rates of nitrogen fixation can be

obtained depending on whether the measurements are extrapolated from laboratory or field-based experiments. Furthermore, the temporal and spatial variability of these processes means that repeated measurements over many different environments is required to get a truly representative estimation of these processes across time and space.

The use of stable isotopes offers a potential solution to the problems described above using the traditional mass balance

approach and newer geochemical inferences. Additionally, the comparison between this approach and those mentioned previously can also provide an indication on the applicability of extrapolated rates. Stable isotopes have been previously used to reconstruct the history of marine nitrogen cycling across large timescales (Altabet, 2007). Work by Mahaffey et al. (2003) used a two-source mixing model to estimate that nitrogen fixation was contributing ~63 % of the nitrogen input to an oligotrophic part of the eastern North Atlantic Ocean. However, such a model represents a simplified version of the nitrogen

cycle and potentially limits its usefulness in assessing the contribution of different processes to the overall nitrogen cycle in all marine environments. This is because systems seldom have just two major processes influencing the isotopic signature of the end point of interest. A more complex system of nitrogen sources was considered by the Korth et al. (2014) study carried out in the Baltic Sea, however, not including any loading terms for the relevant processes neglects the fact that one of the processes will likely have a more significant impact on the cycling of nitrogen than the others will. Therefore, the inclusion





of such loading terms by Altabet and Francois (1994) and Karl et al. (1997) amongst others represents a more complete isotope balance to constrain the cycling of nitrogen in marine environments.

However, studies such as those described above have been carried out over very large spatial scales, and therefore are of limited use in constraining the cycling of nitrogen and hence demonstrating the importance of nitrogen fixation in a local/regional context. Additionally, whilst combined isotope and mass balances have been carried out on smaller spatial scales and have shown that nitrogen fixation makes a significant contribution to the total pool of available nitrogen in these systems (Liu et al., 1996; Radtke et al., 2012; Voss et al., 2005; Woodland and Cook, 2014), the nitrogen-replete status of the sites are not truly representative of marine system worldwide. Consequently, it is important that additional studies that investigate nitrogen cycling are extended to nitrogen-poor coastal marine environments. This is because the availability of nitrogen through processes such as denitrification and nitrogen fixation can exert a large control over primary productivity (Elser et al., 2007), and that coastal margins are often characterised by the presence of large expanses of benthic macrophytes such as seagrass (Charpy-Roubaud and Sournia, 1990), which can also affect the way in which nutrients such as nitrogen are cycled (Orth et al., 2006).

Therefore, this study aimed to:

1. Develop and evaluate the validity of a combined mass and stable isotope balance that describes the isotopic signature of nitrogen in sediment across a nitrogen-poor, temperate, intertidal embayment characterised by large expanses of seagrass.
2. Evaluate the agreement between the rate of nitrogen fixation estimated from the sediment isotope balance and from the extrapolation of discrete measurements.

## 2 Methodology

### 2.1 Study site and isotope balance development

The combined mass and isotope balance described in this study was undertaken across Western Port, Victoria, Australia. It is a nitrogen-poor, tidally influenced shallow marine embayment of ~650 $km^2$ and is roughly 55 km south-east of Melbourne. The major tributaries flowing into the bay are Bass River, Bunyip River, Lang Lang River, Toomuc Creek and Watsons Creek, and there are two separate openings to the ocean located on either side of Phillip Island (Fig. 1; Table S1).

Many possible sources and sinks for nitrogen exist within the marine environment. Past studies have all acknowledged the importance of riverine inputs, atmospheric deposition and nitrogen fixation as being the major sources, whilst sediment burial and denitrification were the major sinks (Brandes and Devol, 2002). This study also considered vegetative assimilation




as an additional sink term, which is commonly overlooked in many mass and isotope balances. Such a term is particularly important in environments such as Western Port, which contain large expanses of vegetation such as seagrass and mangroves. All loadings and isotopic signatures of the end-members/fractionation factors used in this study are listed in Table 1. Whilst most of the data was sourced from previous studies around Western Port and surrounding areas, in some

instances no such data was available and global average values were used.  The development of the isotope and mass balance will be detailed below, whilst a discussion of the source and sink terms used in this balance will be carried out below.

The combined nitrogen mass and isotope balance used in this study is based upon the assumption that the export of nitrogen to the ocean is balanced by inputs resulting from nitrogen fixation, atmospheric deposition and terrestrial run-off. Therefore,

Western Port is considered to be acting in a *pseudo*-closed manner with respect to the cycling of nitrogen, this is despite the fact that there is the potential for considerable tidal exchange of material within the bay. As a consequence, no explicit oceanic term will be considered in the model proposed in this study. The validity of such an assumption will be further discussed along with all other assumptions that underpin the development of the proposed model. Consequently, both the flux of nitrogen and the overall isotopic signature can be considered to be fixed (Freudenthal et al., 2001), resulting in Eq.

15   (1).

$$m_{Sources} \times \delta^{15}N_{Sources} = m_{Sinks} \times \delta^{15}N_{Sinks} \tag{1}$$

As has been discussed previously, the sources of nitrogen that were considered in this model are nitrogen fixation,

atmospheric deposition and riverine inputs. Consequently, the left-hand side of Eq. (1) can be re-written as follows:

$$m_{Sources} \times \delta^{15}N_{Sources} = m_{N\text{-}Fix} \times \delta^{15}N_{N\text{-}Fix} + m_{Riv\text{-}TN} \times \delta^{15}N_{Riv\text{-}TN} + m_{AD\text{-}NH_4^+} \times \delta^{15}N_{AD\text{-}NH_4^+}$$
$$+ m_{AD\text{-}NO_X} \times \delta^{15}N_{AD\text{-}NO_X} \tag{2}$$

Similarly, the right-hand side of Eq. (2) can be re-written to take into account the sink terms that were considered during this study (sedimentary denitrification, sediment burial and assimilation; Eq. (3)).

$$m_{Sinks} \times \delta^{15}N_{Sinks} = m_{Sed} \times \delta^{15}N_{Sed} + m_{Denit} \times \epsilon^{15}N_{Denit} + m_{Ass} \times \epsilon^{15}N_{Ass} \tag{3}$$

Therefore, after combining Eq. (2) and Eq. (3), and rearranging an equation for the isotopic signature of the sediment can be obtained (Eq. (4)).




$$\delta^{15}N_{Sed} = \frac{\left(m_{Sources} \times \delta^{15}N_{Sources}\right) - \left(m_{Denit} \times \epsilon^{15}N_{Denit} + m_{Ass} \times \epsilon^{15}N_{Ass}\right)}{m_{Sed}} \tag{4}$$

## 2.2 Isotope mass balance – sources

### 2.2.1 Atmospheric Deposition

Long-term rainfall data for the period 2006–2016 was obtained for Melbourne and was then extrapolated to the entire area of
Western Port (http://www.melbournewater.com.au/waterdata/Pages/waterdata.aspx, February 2017). Based upon the
previous work of Lansdown (2009) and Wong et al. (2014) it was assumed that the concentrations of $NH_4^+$ and $NO_X$ in the
rainfall was $0.2 \pm 0.02$ mg N $L^{-1}$ and $0.204 \pm 0.01$ mg N $L^{-1}$ respectively, resulting in an estimated total input of ~$186 \pm 20$ t
N $yr^{-1}$. Similarly, it was assumed that the end-members for the atmospheric deposition of $NH_4^+$ and $NO_X$ were $1.0 \pm 2.0$ ‰
and $-1.4 \pm 2.9$ ‰ respectively based upon the previous work of Lansdown (2009) and Wong et al. (2014).

### 2.2.2 Nitrogen Fixation

The production of bioavailable nitrogen through nitrogen fixation was estimated to be $434 \pm 40$ t N $yr^{-1}$ based upon work by
Russell et al. (2016). This value was obtained by extrapolating rates from vegetated and non-vegetated core incubations
collected at Corinella, Coronet Bay and Rhyll (Fig. 1) throughout the austral winter-summer period (July 2014–March
2015). Based upon the available literature it was assumed that the isotopic end-member for nitrogen fixation was $0.0 \pm 1.0$ ‰
(Owens, 1988).

### 2.2.3 Riverine Inputs

The catchments and tributaries that surround Western Port also make a significant contribution of nitrogen input to the bay,
with nitrogen entering in various forms including particulate and dissolved nitrogen (in particular nitrate, $NO_3^-$). Potential
contributions of dissolved organic nitrogen were neglected because riverine organic material generally nitrogen poor
(Hedges et al., 1997), and terrestrially-derived organic material makes a very low contribution to the overall pool in marine
environments (Opsahl and Benner, 1997). It was estimated that yearly input of total nitrogen was ~$610 \pm 30$ t N $yr^{-1}$, with an
overall isotopic signature of ~$9.2 \pm 2.8$ ‰. This estimate was made using information available from the Victorian Water
Measurement Information System (http://data.water.vic.gov.au/monitoring, January 2017). Nutrient concentrations for both
total nitrogen (TN) and ($NO_3^-$) in addition to flow data for the major tributaries was available for the period 1990–2013.
Loading calculations were made using the Generator Uncertainty Measures and Load Estimates using the Alternative
Formulae program (GUMLEAF version 0.1 alpha; Tan et al., 2005) in conjunction with the flow regime-stratified Kendall's
ratio estimator (Tan et al., 2005).





Additional water samples were collected from the main tributaries flowing into Western Port between April 2014 and May 2015 in order to determine the isotopic signature ($\delta^{15}N$) of the catchment-derived $NO_3^-$ flowing into the bay. These samples were collected from the most downstream freshwater section of each river/creek (Table S1), to ensure that all samples collected were terrestrially-derived and not influenced by tidal exchange. Samples were filtered through 0.22 µm Sartorius

Minisart syringe filters and frozen until analysis. The approach of McIlvin and Altabet (2005) was used to determine the isotopic signature of $^{15}N$-$NO_3^-$. Initially, cadmium was used to reduce $NO_3^-$ to $NO_2^-$, afterwards sodium azide in an acetic acid buffer was used to further reduce the $NO_2^-$ to $N_2O$. The $N_2O$ produced was analysed on a Hydra 20-22 continuous flow isotope ratio mass spectrometer (CF-IRMS; Sercon Ltd., UK) interfaced to a cryoprep system (Sercon Ltd., UK), the precision of the stable isotope analysis was ± 0.3 ‰ (SD; $n$=5).

The isotopic signature of the total nitrogen load was estimated using the loading weighted average isotopic signature of particulate (SPN) and dissolved ($NO_3^-$) nitrogen. These samples of SPN ($n$=30; November 2015–October 2016) and $NO_3^-$ ($n$=21; April 204–May 2015) were collected under high and low flow conditions to ensure that the riverine isotopic signature estimated was representative of the annual input of riverine nitrogen into Western Port. For the purposes of this calculation it

was assumed that the loading of particulate nitrogen was the difference between the total and $NO_3^-$ loadings. Pre-ashed Whatmann 25 mm GF/F paper was used to determine the isotopic signature of the particulate nitrogen. These filters were then dried to a constant weight at 60 °C for 48 hours before being analysed using an ANCA GSL2 elemental analyser interfaced to a Hydra 20-22 continuous flow isotope ratio mass-spectrometer (Sercon Ltd., UK). Stable isotope data for nitrogen was reported in delta notation (Eq. (5)) relative to the $^{15}N/^{14}N$ ratio of air, the precision of the stable isotope analysis

was ± 0.2 ‰ (SD; $n$=5) and the precision of the elemental analysis ± 0.5 µg (SD; $n$=5).

$$\delta^{15}N = \left( \frac{\left(^{15}N/^{14}N\right)_{sample}}{\left(^{15}N/^{14}N\right)_{N_2\text{-Air}}} - 1 \right) \times 1000 \qquad (5)$$

Based upon a loading of ~220 ± 30 t N $yr^{-1}$ and an isotopic signature of ~12.6 ± 7.0 ‰ for riverine $NO_3^-$ and a loading of

~390 ± 40 t N $yr^{-1}$ and an isotopic signature of ~7.3 ± 1.4 ‰ for particulate nitrogen, it was estimated the overall isotopic signature of the total riverine nitrogen input into Western Port was ~9.2 ± 2.8 ‰ (Eq. (6)).

$$\delta^{15}N_{Riverine\ TN} = \frac{m_{Riverine\ NO_X}}{m_{Riverine\ TN}} \times \delta^{15}N_{Riverine\ NO_X} + \frac{m_{Riverine\ SPN}}{m_{Riverine\ TN}} \times \delta^{15}N_{Riverine\ SPN} \qquad (6)$$



### 2.3 Isotope mass balance – sinks

#### 2.3.1 Denitrification

In earlier work by Brandes and Devol (2002), the contribution of denitrification to nitrogen isotope and mass balances was separated into sedimentary and water column derived denitrification. However, as the concentration of dissolved inorganic

nitrogen in the water column was extremely low (Russell et al., 2016), and no anoxia has ever been recorded in the well mixed water column, it was assumed that that water column derived denitrification represented an insignificant sink and was ignored. Prior core incubation experiments by Russell et al. (2016) estimated that the yearly removal of bioavailable nitrogen through denitrification was $228 \pm 10$ t N yr$^{-1}$ across the entirety of Western Port. In this study, the isotopic value arising from sedimentary denitrification was assumed to be $3.5 \pm 2.0$ ‰ based upon the global estimates of Brandes and Devol (2002).

The use of a globally estimated fractionation factor for sedimentary denitrification is similar to those that have been determined on smaller spatial scales, with Alkhatib et al. (2012) calculating a fractionation factor of $4.6 \pm 2$ ‰ along the St Lawrence Estuary in Canada.

#### 2.3.2 Sediment Burial

Estimates for the burial of sediment within Western Port have been reported to vary between 70–100 kt sediment yr$^{-1}$, with

the deposition rate of sediment around Corinella being found to have increased rapidly over the past 40 years (Hancock et al., 2001). Consequently, a maximal sediment burial rate of 100 kt yr$^{-1}$ was assumed, and based on elemental analysis of the sediment (~0.5 % wt nitrogen), it was estimated that the rate of nitrogen burial through sediment accretion was 500 t N yr$^{-1}$. It was assumed that the 30 kt yr$^{-1}$ range in burial rates estimated by Hancock et al. (2001) represented the maximum range of likely burial rates, which in turn meant that 99 % of the measurements (3 standard deviations) would fall within this range.

Consequently, it was calculated that one standard deviation of this range would be 25 t N yr$^{-1}$ (based on ~0.5 % wt nitrogen). To determine the average actual isotopic value of sediment nitrogen, against which the model output would be validated, intact sediment samples were collected from a variety of sites in Western Port (Fig. 2) using Perspex tubes to a depth of ~5 cm. These samples were dried to a constant weight at 60 °C for 48 hours and then homogenised using a mortar and pestle. Up to 135 mg of sample was used for analysis of sediment nitrogen content and isotopic analysis, using the CF-IRMS

described above (*n*=93; February 2012–July 2016).

#### 2.3.3 Assimilation

It was assumed that the contribution of assimilation to the removal of nitrogen within Western Port represented the balance of the nitrogen sources, and additional sinks such as sediment burial and denitrification. The imposition of this condition was necessary so that the underlying assumption of the *pseudo* closed system cycling of nitrogen within the bay that underpins

our nitrogen isotope and mass balance held true. This validity of such an assumption will be further discussed in the discussion. Consequently, it was estimated that assimilation resulted in the removal of ~500 t N yr$^{-1}$ through a combination





of vegetative, phytoplankton and microphytobenthic assimilation. It was assumed that the overall isotopic fractionation factor for assimilation would be the average fractionation factor for vegetative assimilation and phytoplankton/MPB assimilation. In nitrogen-poor aquatic environments, it has been concluded that there is little to no fractionation of dissolved inorganic nitrogen (Finlay and Kendall, 2007). Needoba et al. (2003) and Altabet (2001) reported that the fractionation of

nitrogen during assimilation by phytoplankton varied from 0.9 to 23 ‰, with the majority of the fractionation factors where in the range of 2–8 ‰. Consequently, the average of the likely fractionation factor for vegetative assimilation (~0 ‰), and the upper range for phytoplanktonic assimilation (~8 ‰) returned a fractionation factor of 4 ‰, which was subsequently used in this study. The error associated with the fractionation factor of nitrogen assimilation (1 ‰) was estimated to be the average of the fractionation factors for vegetative assimilation (0 ‰; Russell et al. in review) and phytoplanktonic

assimilation (~2 ‰).

### 2.3.4 Potential oceanic contribution

Due to the intertidal nature of Western Port, the ocean is able to act as both a source and a sink of nitrogen to the sediment and therefore could influence the isotopic signature of the sediment. For the purposes of this isotope balance, it was assumed that any contribution of oceanic sources to the isotopic signature would end up being effectively combined with those

derived from the catchment (riverine TN). Such an assumption is based upon the fact that the predominant circulation of water within Western Port is clockwise in direction. At the confluence zone located north of Cowes (Fig. 1), the circulating waterbody normally splits in two, with a high proportion heading north around French Island, and the rest heading past Rhyll down to the eastern entrance to the bay Harris et al. (1979). Compared to other parts of the bay, the circulation of water clockwise is considered to be relatively slow, and it is in this section of the bay, particularly north-east of French Island that

the majority of the catchments flow out into Western Port. This would allow mixing of the catchment-derived inputs and the incoming oceanic water, thereby resulting in a combined oceanic-riverine input of nitrogen to the sediment. Samples of particulate nitrogen were obtained from Cowes and San Remo during the incoming and outgoing tides (Table S1) using pre-ashed Whatmann 25 mm GF/F paper ($n$=43–47; October 2015–January 2017). These filters were processed and analysed using the CF-IRMS described previously. With extremely low concentrations of $NH_4^+$/$NO_X$ (<1 μM), and the small fraction

of dissolved organic nitrogen as a proportion of total marine-fixed nitrogen B&D 2002, it was assumed that particulate nitrogen was the dominant source oceanic nitrogen. On average, it was determined that the isotopic signature of the particulate material was 6.7 ± 1.1 ‰ across both sites for all sampling periods. Furthermore, for the purposes of this isotope balance, an equal contribution of the oceanic (~6.7 ± 1.1 ‰) and riverine sources (~9.2 ± 2.8 ‰) was assumed. This resulted in an average isotopic signature for the combined oceanic/riverine end-member of ~8 ± 1.5 ‰.

### 2.3.5 Data analysis

R 3.2.2 (R Core Team, 2015) was used to estimate the nitrogen isotopic signature of the sediment ($\delta^{15}N_{Sed}$) using a Monte-Carlo type simulation. This simulation was run for 10,000 iterations using a model developed in this study (Fig. S1). Linear





regression analysis investigating the strength of the response of the loading and isotopic end-member/fractionation factors on the isotopic signature of sediment (Fig. S2 and Fig. S3) were carried out in GraphPad Prism 7. The Student's *t*-test was also used to compare the actual sedimentary $\delta^{15}N$ and the estimated value and to investigate the effect of varying the relative contribution of oceanic and terrestrially-derived nitrogen on the estimated sedimentary $\delta^{15}N$.

Differences in isotopic and nitrogen composition of particulate material from Cowes and San Remo were tested for significance using two factor analysis of variance (ANOVA) in R. The TukeyHSD post hoc test was used in the event that significant responses were returned for the two factor ANOVA. Data was evaluated graphically using plots of residuals and boxplots to test for the homogeneity and normality of variances Quinn and Keough (2002). For all analyses $p<0.05$ was
chosen as the level of significance for the rejection of the null hypothesis.

## 3 Results

### 3.1 Model output

Isotopic analysis of sediment from various locations around Western Port showed that on average $\delta^{15}N$ was 3.9 ± 1.2 ‰ (Fig. 3). Little variation in the signature was observed around the majority of the bay, with the highest values centred on the
north of the bay, in closest proximity to the headwaters of the main tributaries. Whereas, the isotopic signature of the sediment that was estimated after 10,000 iterations of the model proposed in the study was 4.1 ± 2.5 ‰ (Fig. 3; Fig. S1).

### 3.2 Validity of assumptions made

The proportional contribution of riverine TN inputs and oceanic inputs were found to greatly influence the isotopic signatures of sediments. Under the scenario where the contribution of riverine TN inputs was assumed to be 100 % (no
oceanic contribution), and all other loadings/end-members were held constant, the isotopic signature of sediment was estimated to be ~5.7 ‰ (Fig. 4). Conversely, if the contribution of oceanic inputs was assumed to be 100 % (no riverine contribution), the estimated isotopic signature of the sediment was estimated to be ~2.6 ‰ (Fig. 4). Both the isotopic signature and the nitrogen content of the particulate nitrogen in the water column was found to remain relatively over the tidal cycle. Samples from Cowes had particulate nitrogen isotopic signatures of between 5.4 and 6.8 ‰ and a nitrogen
content of between 50 and 56 µg N L$^{-1}$ (Fig. 5). Whereas, at San Remo the particulate nitrogen isotopic signatures of between 6.4 and 7.0 ‰, with a nitrogen content of between 29 and 33 µg N L$^{-1}$ (Fig. 5).

### 3.3 Scaling of the nitrogen fixation loading

Due to the uncertainty surrounding the extrapolation of measured nitrogen fixation rates (Karl et al., 2002) to bay-wide loadings, the overall loading was set such that that the estimated and actual sediment $\delta^{15}N$ were equal. It was estimated that a

loading of 452 t N yr$^{-1}$ was required for the output of the model to equal the actual isotopic signature of nitrogen in the sediment (Fig. 6). This compares well with the previous estimate of 434 t N yr$^{-1}$ obtained by the extrapolation of rates from core incubation experiments (Russell et al., 2016). This updated estimate of the nitrogen fixation loading increases its contribution to the total input of nitrogen to Western Port is ~36 %; up from initial estimate of ~35 %.

**3.4 Sensitivity analysis**

Based upon the linear regression analysis of each variable against the estimated isotopic value of the sediment it is clear that there is considerable variability in the quality of the fits obtained (Table 2; Fig. S2 and Fig. S3). It is apparent that the loading terms of both the sinks and the sources did not exert an appreciable control over the isotopic value of the sediment. None of the loading terms individually were able to explain more than 2 % of the variation in the isotopic signature of the

sediment (Table 2), with nitrogen fixation exhibiting the highest total variation of 1.7 % ($R^2$ of ~0.017). In contrast, the isotopic signature of the end-members/fractionation factors for both the source and sink terms were found to exert a much stronger control over the isotopic signature of the sediment. The combined riverine/oceanic inputs of TN were able to explain almost 50 % of the variation experienced in the sediment $\delta^{15}$N (Table 2). Whilst the assimilation, denitrification and nitrogen fixation end-members were able to explain 14.1 %, 13.5 % and 11.7 % respectively of the variation experienced in

the sediment $\delta^{15}$N (Table 2).

**4 Discussion**

How nitrogen is cycled in marine environments is crucially important, with its availability a significant control on the primary productivity of such environments (Elser et al., 2007). Debate surrounds the relative importance of these processes owing to methodological issues associated with the measurement of these processes. The development of a combined

nitrogen mass and isotope balance represents a novel solution to this problem. The output and validity of the assumptions made during the development of the model proposed in this study will be initially discussed. Following this, the controls on the isotopic signature of the sediment will be investigated. Finally, the agreement between the bay-wide loading of nitrogen fixation predicted by this model and from previous studies will be assessed.

**4.1 Model output**

The excellent agreement between the estimated (4.1 ± 2.5 ‰) and measured (3.9 ± 1.2 ‰) sediment nitrogen isotopic values indicates that the model produced in this study is able to accurately describe the cycling of nitrogen in Western Port. Such an outcome leads further weight to the ability of combined nitrogen isotope and mass balances to be able to accurately describe nitrogen cycling in marine environments, such as in the Baltic Sea carried out by (Radtke et al., 2012). Again, in this study the validity of the proposed model was assessed by the degree of agreement between the estimated and actual sedimentary

nitrogen isotopic signature.



Stable isotopes on their own have been used to provide an indicative assessment of the dominant processes occurring in marine environments. Cremonese et al. (2013) suggest that sediment isotopic signatures of between 2–6 ‰ indicate the co-occurrence of denitrification and nitrogen fixation. It also follows that sediments with isotopic values towards the lower end of the range are more dominated by nitrogen fixation, whereas sediment with isotopic values at the higher end of the range are dominated by denitrification. With the sediment isotopic signature of 3.9 ‰ measured in Western Port representing an intermediate value relative to the range quoted by Cremonese et al. (2013), this is indicating that there is a co-occurrence of nitrogen fixation and denitrification within Western Port. Such findings are consistent with previous studies in Western Port that confirmed using direct measurements the co-occurrence of both processes for range of vegetated and non-vegetated sediments within the bay (Russell et al., 2016).

## 4.2 Validity of assumptions made

The development of the mass and isotope balance model was underpinned by the assumption that Western Port was acting as a closed system respect to the cycling of nitrogen. Over the course of the incoming and outgoing tides at both entrances to the bay no change in the nitrogen content of the particulate material or its nitrogen isotopic signature was detected. This clearly demonstrates that the bay experienced no net import or export of nitrogen, therefore, the apparent deficit in nitrogen sinks must be as a result of assimilation. Previous estimates of net primary production of seagrass in Western Port have been in the order of 5050 t C yr$^{-1}$ (Clough and Attiwill, 1980), which is equivalent to ~281 t N yr$^{-1}$ based on a Redfield-type C:N ratio of 18:1 for macrophytes (Atkinson and Smith, 1983). It has also been estimated that macrophytes and long-lived macroalage account for ~40 % of productivity in shallow coastal systems (Charpy-Roubaud and Sournia, 1990). This, therefore, suggests that the total net primary productivity of Western Port is in the order of ~700 t N yr$^{-1}$. If a large proportion of this met through assimilation, then our assumption that assimilation represents an annual nitrogen sink of ~531 t N yr$^{-1}$ is indeed reasonable. If one considers the ability of seagrass to meet ~20 % of its annual nitrogen requirements for primary production through internal recycling (Hemminga et al., 1991), this leaves ~562 t N yr$^{-1}$ needing to be assimilated, which is in very close agreement with our assumption.

The combining of the oceanic and riverine sources into a single term for the purposes of this nitrogen isotope and mass balance is reasonable considering how the isotopic signature of particulate nitrogen was observed to change during high and low tides. No statistically significant difference in the isotopic signature of the particulate nitrogen was observed at either Cowes or San Remo from low tide to high tide ($p > 0.05$; Table S2). If the riverine inputs of nitrogen made no contribution to the sediment isotopic signature, it would be expected that an increase in the isotopic signature of the particulates in the outgoing tide would occur. This is because the input of catchment-derived nitrogen (TN) has an isotopic signature ~2.5 ‰ higher than the oceanic inputs. Furthermore, when the nitrogen balance was carried out assuming that the riverine inputs did not contribute to the sediment isotopic signature, the estimated isotopic signature was ~2.6 ‰ (Fig. 4), which is clearly lower



than the actual value of 3.9 ‰. A similar argument can be made for why it's highly unlikely that the oceanic inputs of nitrogen (SPN) would also make no contribution to the sediment isotopic signature. In this case, the riverine inputs would represent a 100 % contribution, and should increase the isotopic signature to ~5.7 ‰ (Fig. 5), a value that is appreciably higher than the sediment isotopic signature of 3.9 ‰ that is observed in reality. Therefore, the combining of the oceanic and

riverine sources into a single source term appears to be justified in this case.

### 4.3 Sensitivity analysis

The control that the isotopic end-members and fractionation factors have over the output of our model describing $\delta^{15}N_{Sed}$ is consistent with previous research. Both Altabet (2006) and Mahaffey et al. (2005) state that it is the choice of the end-members and fractionation factors that will have the biggest influence on $\delta^{15}N$ derived from stable isotope balances or

budgets. This is consistent with our linear regression analysis that found that all terms excluding atmospheric deposition could individually explain >10 % of the variation in isotopic signature of the sediment (Table 2). The particularly strong influence of the combined oceanic/riverine term in this regression analysis is unsurprising when one considers just how isotopically heavy this end-member is and how well constrained relative to all other end-members/fractionation factors it is. Whilst similar arguments could be made for the influence of the atmospheric deposition terms on the sediment isotopic

signature, it must be noted that the relative errors associated with these terms is quite high. Therefore, it is likely that any underlying influence on the sediment isotopic signature will be masked by the relatively high variability of these end-members. Previously Altabet (2006) has suggested that overall the fractionation factors associated with removal processes such as denitrification should normally make the biggest contribution as a result of significant fractionation during these processes. Such conclusions are based upon the presence of large fractionation factors of ~20 ‰ that are associated with

water column-derived denitrification (Brandes and Devol, 2002). However, in the absence of water column-derived denitrification, and the limited fractionation associated with sedimentary denitrification (3.5 ‰), it is unsurprising that denitrification on its own isn't the largest contributor to the sediment isotopic signature.

It would also be expected that the nitrogen flux of both the source and sink terms in this model would have either a positive

or negative effect on the sedimentary isotopic signature. Work by Altabet (2007) suggests that this isn't necessarily the case. For instance, no change in $\delta^{15}N$ was observed with a ± 30 % change in the amount of sedimentary denitrification, but changes of >2 ‰ where observed for similar changes in water column denitrification and nitrogen fixation. However, these scenarios where undertaken for a system that not only were not in a steady state, but also for an unbalanced nitrogen flux. Consequently, it is likely that our imposition of steady state and a balanced nitrogen flux resulted in the effective cancelling

out of the effect of the flux terms, thereby resulting in no net effect on $\delta^{15}N$. However, as the validity of our assumption of a closed system appears reasonable, it is unsurprising that the end-members and fractionation factors exert a more significant control over sediment $\delta^{15}N$ than the loading terms do for Western Port.





## 4.4 Scaling of the nitrogen fixation loading

One of the common criticisms of extrapolating loadings and rates from discrete measurements across large temporal and spatial scales is the fact that such experiments will likely only provide a snap-shot of the true rates over short timescales (Mahaffey et al., 2005). It has also been suggested that the use of the acetylene reduction technique as a method to measure nitrogen fixation is problematic owing to uncertainty over the appropriate acetylene reduction to nitrogen fixation stoichiometry (Karl et al., 2002). Furthermore, intact core incubations may also result in an under-estimation of the true rate of nitrogen fixation owing to the inability to attain saturating acetylene concentrations deep within the sediment (Capone, 1988). However, the extremely good agreement (4 % difference) between the two methods utilised to estimate the bay-wide nitrogen fixation loading suggests that such errors were unlikely to have significantly affected the results of the initial extrapolations. Therefore, it can be concluded with reasonable confidence that the nitrogen fixation loading in Western Port is between 434 to 452 t N yr$^{-1}$.

In many previous studies the importance of nitrogen fixation to the total inputs of various marine studies has been found to vary from between 3 and 40 % (Howarth et al., 1988; Huber, 1986; Larsson et al., 2001; Radtke et al., 2012; Russell et al., 2016; Woodland and Cook, 2014). The studies that found the highest contribution of nitrogen fixation to the total input of nitrogen typically had cyanobacterial blooms or cyanobacterial mats present. Higher contributions of between $37.9 \pm 5.1$ % to $58.8 \pm 2$ % were reported in the study by Korth et al. (2014), which used an isotopic mixing model to estimate the contribution of nitrogen fixation throughout the Baltic Sea. However, in the construction of this isotope mixing model isotopic effects of denitrification and assimilation on the pool of $NO_3^-$ were neglected. This is potentially problematic though, with Altabet (2006) stating that in the absence of fractionation effects due to removal processes, the oceanic $\delta^{15}N$ in the absence of heavily enriched inputs and dominated by nitrogen fixation, would be isotopically lighter than the average oceanic value of ~5 ‰.

With respect to the contribution of nitrogen fixation to the total input of nitrogen in Western Port, this study has estimated that 35–36 % of the inputs are derived from nitrogen fixation. Such estimates are slightly lower the estimates of Russell et al. (2016) in Western Port, however, this slight discrepancy is likely due to the different estimates of the input to rainfall-derived inputs of nitrogen to the bay. Previously it had been assumed that $NO_X$ was the dominant form of nitrogen (Codispoti et al., 2001), whilst this study explicitly accounted for both $NH_4^+$ and $NO_X$. Nonetheless, despite different approaches to accounting for the rainfall-derived input of nitrogen, it can be concluded that nitrogen fixation represents a substantial portion of the total inputs of nitrogen to Western Port. Whilst these estimates are at the higher end of those reported in previous studies, there was no evidence for the presence of widespread cyanobacterial blooms and mats. Previous work by Russell et al. (2016) has shown that this nitrogen fixation was occurring as a result of both sulfate-reducing bacteria and 'freely-associated' cyanobacterial assemblages. This suggests that the such high rates of nitrogen fixation relative to

other inputs of nitrogen are a direct result of the nitrogen-poor conditions that exist within Western Port. Not only has it been shown previously that the catchment-derived inputs of nitrogen to Western Port are low on both a local and global scale (Woodland et al., 2015), there is also evidence for nitrogen limitation in seagrass meadows found in Western Port (Russell et al., 2016).

## 5 Conclusion

This paper has described the development of a combined mass and isotope balance to help investigate the cycling of nitrogen in Western Port, a nitrogen-poor, temperate, intertidal embayment in south-east Australia. To test the validity of the proposed model, $\delta^{15}N_{sed}$ was calculated and compared to the average actual isotopic signature of nitrogen in sediment throughout Western Port. The model estimated that $\delta^{15}N_{sed}$ was 4.1 ± 2.5 ‰, compared to the actual value of 3.9 ± 1.2 ‰, this indicates that the proposed model accurately accounts for the cycling of nitrogen within Western Port. Sensitivity analysis of each source and sink term confirmed that it was the end-member/fractionation factors of the combined riverine/oceanic inputs, nitrogen fixation, assimilation and sedimentary denitrification that exerted the strongest control over $\delta^{15}N_{sed}$. It was estimated that a 4 % increase in the loading of nitrogen fixation to 452 t N yr$^{-1}$ was required to achieve parity between the actual and estimated sediment isotopic signature. This indicates that approximately 36 % of the total nitrogen inputs to Western Port were derived from nitrogen fixation, which is in close agreement with previous studies, despite potential issues associated with extrapolation of discrete measurements over larger spatial and temporal scales. This study serves to further highlight the importance of nitrogen fixation to coastal marine environments that are characterised by nitrogen limited conditions and large expanses of seagrass.

**Author Contribution**

All authors contributed to the design, undertaking experiments, data analysis and interpretation, and the preparation of the manuscript.

**Competing Interests**

The authors declare that they have no conflict of interest.

**Data Availability**

The raw data is available upon request by contacting the corresponding author.



## Acknowledgements

This work has been funded by the Australian Research Council (LP130100684), Melbourne Water, Parks Victoria and the Victorian Environmental Protection Authority. Douglas Russell was supported by an Australian Government Research Training Program Scholarship (RTP). The authors wish to thank Andy Longmore and Dr. Todd Scicluna for their collection

of the sediment and oceanic samples respectively and Dr. Victor Evrard for the initial isotopic analysis of the sediment samples.

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





**Table 1: Summary of loading and isotopic end-members/fractionation factors used in this study.**

| Sources | Average Loading (t N/yr ± SD) | Reference | Average End-Member / Fractionation Factor (‰ ± SD) | Reference |
|---|---|---|---|---|
| Riverine TN | 612 ± 30 | This Study | 9.2 ± 2.8 | This Study |
| Nitrogen Fixation | 434 ± 40 | Russell et al. 2016 | 0.0 ± 1.0 | Owens 1988 |
| Rainfall - $NH_4^+$ | 92 ± 13 | This Study | 1.0 ± 2.0 | Lansdown 2009 |
| Rainfall - $NO_X$ | 94 ± 9 | This Study | -1.4 ± 2.9 | Lansdown 2009, Wong et al. 2014 |
| Oceanic | N/A | N/A | 6.7 ± 1.1 | This Study |
| **Sinks** | | | | |
| Sediment Burial | 500 ± 50 | This Study, Hancock et al. 2001 | N/A | N/A |
| Denitrification | 228 ± 10 | Russell et al. 2016 | 3.5 ± 2.0 | Brandes and Devol 2002 |
| Assimilation | 503 ± 58 | This Study | 4.0 ± 1.0 | Altabet 2001, Needoba et al. 2003, Russell et al. in review |
| Combined Oceanic and Riverine Input | N/A | N/A | 8.0 ± 1.5 | This Study |



**Table 2: Summary of linear regression analysis.**

| Variable | Regression $R^2$ |
| --- | --- |
| Riverine TN Loading | 0.006 |
| Nitrogen Fixation Loading | 0.017 |
| Atmospheric Deposition $NH_4^+$ Loading | <0.001 |
| Atmospheric Deposition $NO_X$ Loading | <0.001 |
| Denitrification Loading | <0.001 |
| Sediment Burial Loading | <0.001 |
| Assimilation Loading | 0.003 |
| Riverine TN End-member | 0.497 |
| Nitrogen Fixation End-member | 0.117 |
| Atmospheric Deposition $NH_4^+$ End-member | 0.021 |
| Atmospheric Deposition $NO_X$ End-member | 0.047 |
| Denitrification Fractionation Factor | 0.135 |
| Assimilation Fractionation Factor | 0.141 |





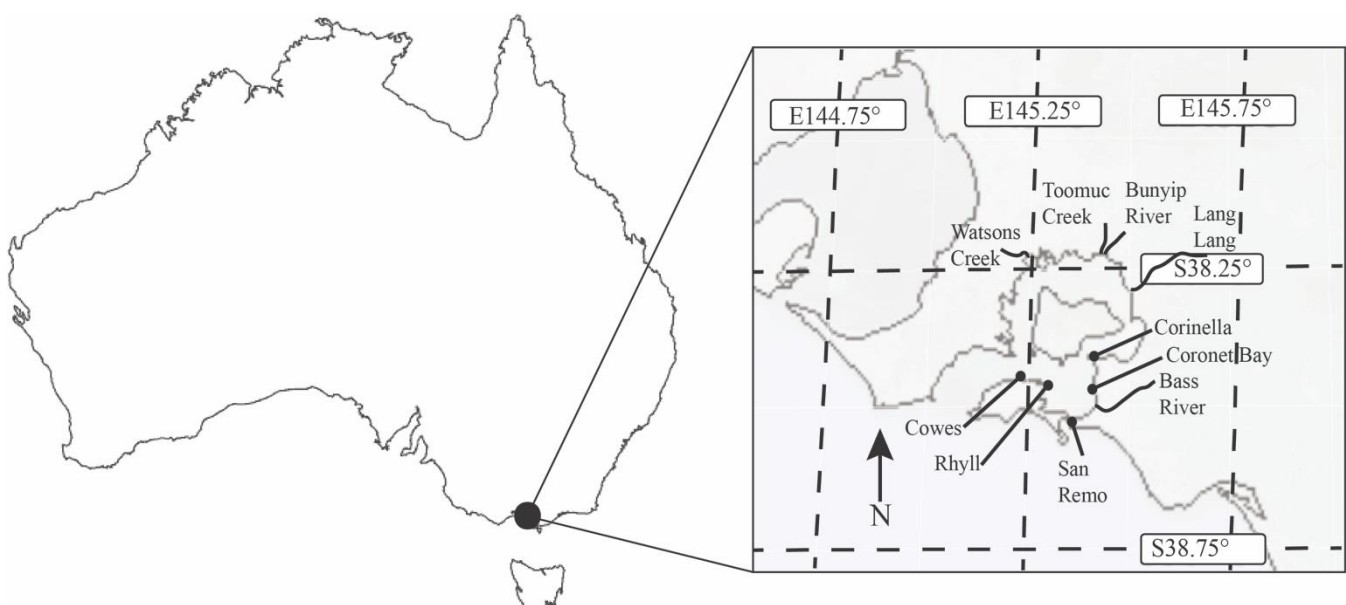

**Figure 1: Location of study sites in Western Port, Victoria, Australia.**





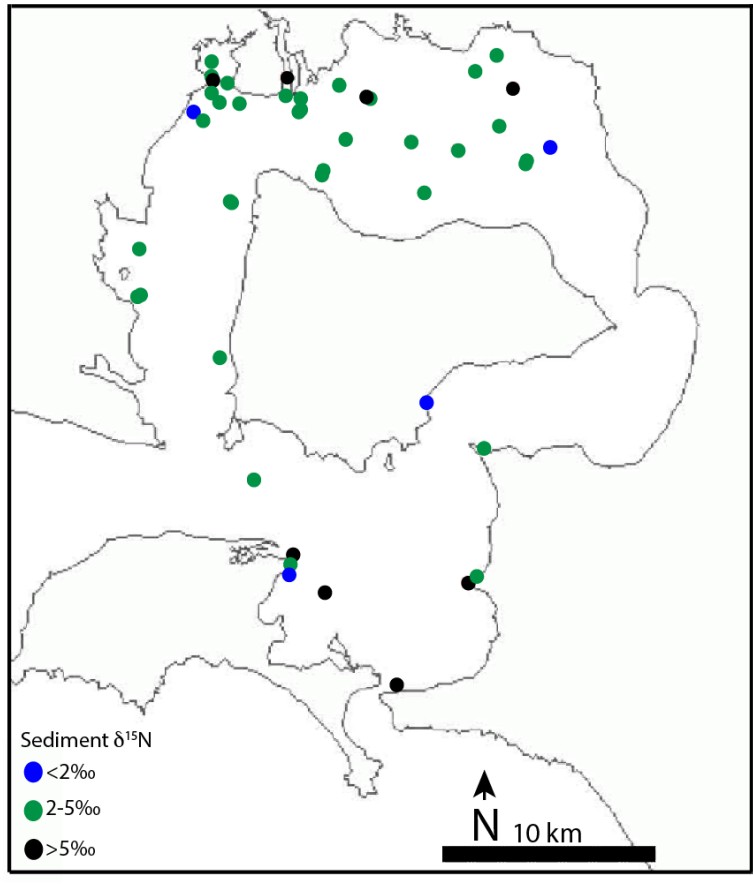

**Figure 2: Location of discrete sediment samples taken from Western Port, Victoria, Australia. The coloured dots represent the isotopic signature of the sediment ($\delta^{15}$N).**




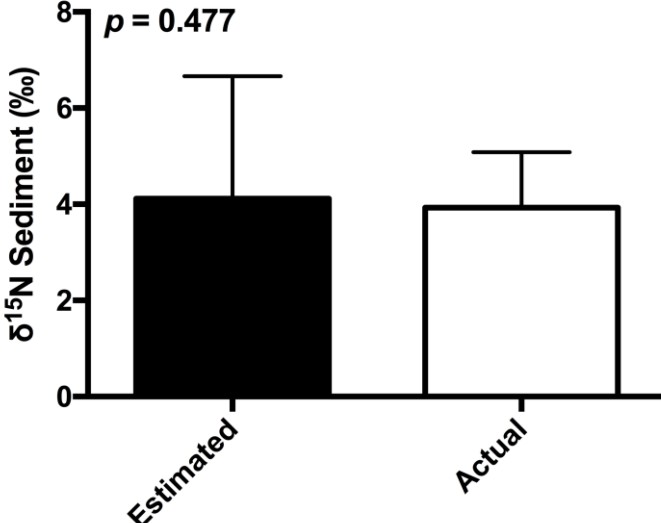

**Figure 3: Estimated vs. actual isotopic signature ($\delta^{15}$N) of sediment in Western Port. Isotopic signature mean ± S.D.**





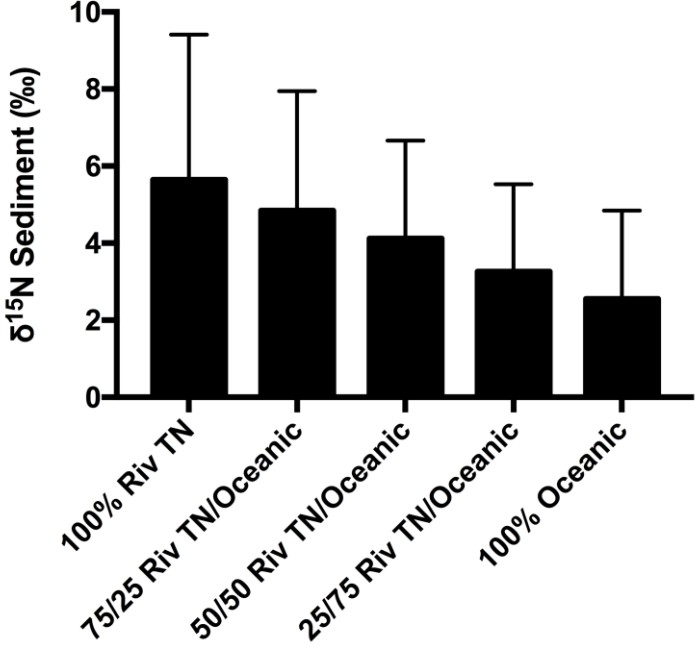

**Figure 4: Comparison of sediment δ$^{15}$N values for different riverine/oceanic input combinations. Isotopic signatures mean ± S.D.**



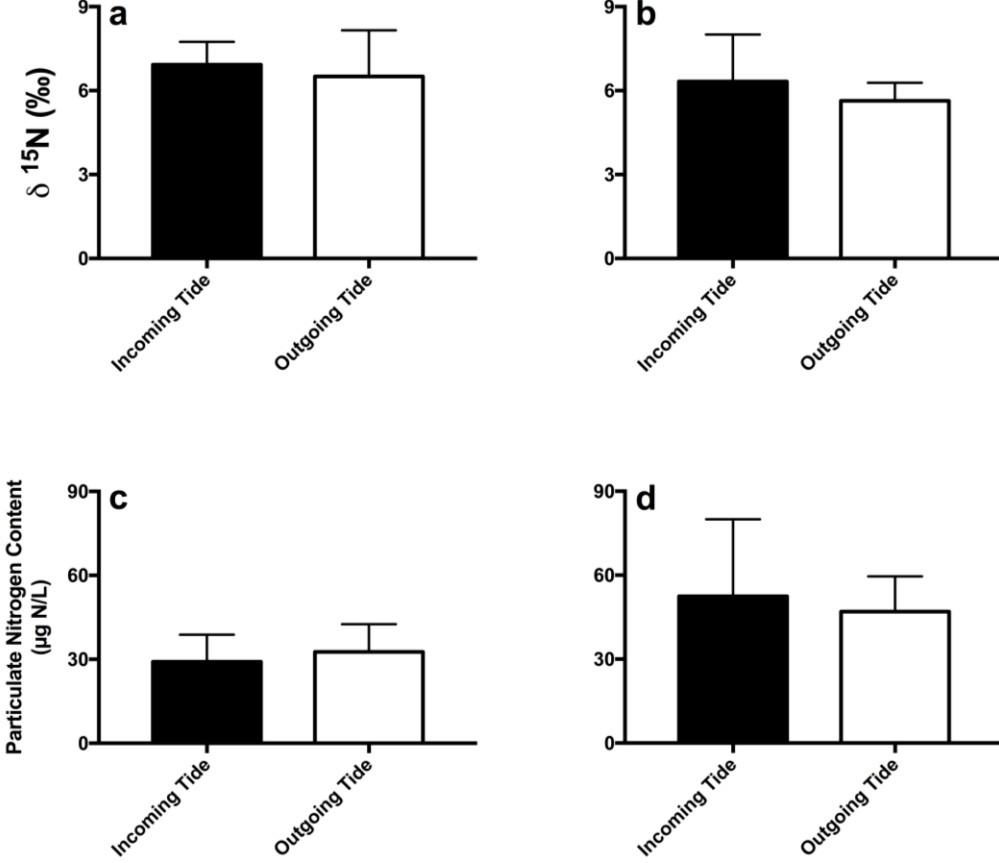

**Figure 5: Comparison of particulate δ$^{15}$N for the incoming and outgoing tides at (a) San Remo and (b) Cowes, and the particulate nitrogen content for the incoming and outgoing tides at (c) San Remo and (d) Cowes. All values are mean ± S.D.**





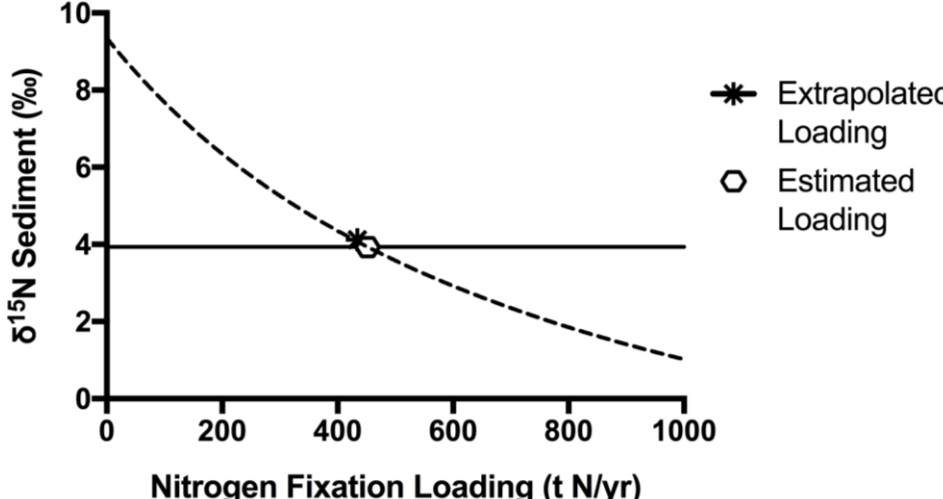

**Figure 6: Scaling of extrapolated nitrogen fixation loadings. The solid black line represents the actual sediment δ¹⁵N, and the dashed line represents the estimated sediment δ¹⁵N for nitrogen fixation loadings of 0–1000 t N yr⁻¹. An estimated loading of 452 t N yr⁻¹ is required to achieve parity between the actual sediment δ¹⁵N and the model output described in this study.**