# Peer review of "The importance of nitrogen fixation to a temperate, intertidal embayment determined using a stable isotope balance mass approach"

_Biogeosciences, 2017_

## Referee Comment (RC1) · Anonymous Referee #1 · 10 Jan 2018

Russel and co-workers have attempted to understand the respective roles of nitrogen fixation and denitrification in an intertidal embayment using the stable isotopes of nitrogen. They estimated that the Western Port receives 36% of the bioavailable nitrogen through nitrogen fixation. Although their findings are interesting, their model requires robustness and better description. As nitrogen cycle is not as simple as presented, authors need to take into account several other end members (such as dissolved organic nitrogen) in their mass balance model. Use of fractionation factor has been completed ignored in their model. My detailed review is as below:

[Figure]

Major comments:

1. This model (equations presented on page 5-6) is very trivial. Nitrogen cycle is very dynamic, many components of the cycle are missing in the model, such as anaerobic ammonium oxidation that occurs in sediments (Thamdrup & Dalsgaard, 2002) and flux of the dissolved organic nitrogen (Jickells et al., 2017) to name a few. Moreover, isotopic fractionation factors ($\varepsilon$) for these processes are different – not much is known about their precise values. None of the equations presented here have considered fractionation factors barring equation (3), and that too has inappropriately incorporated fractional factors. In addition, each term (expression) used in equations must be defined clearly, for example, what does mSources in equation (1) stand for? In a nutshell, equations need elaboration and model requires robustness.

2. These (model) equations are equations are time independent. We understand that biogeochemical processes are time dependent – for example N2 fixation is more in some season, while denitrification would dominate in some other season. So how good these isotope mass balance equation can represent such processes?

3. It is stated that equations (2) and (3) provide equation (4) (Page 5, line 30) but I guess equations (1) and (3) provide (4).

4. Section 2.2.1 on atmospheric deposition. Dissolved organic nitrogen in an important source of nitrogen that has been ignored.

5. In the same section, how was atmospheric flux estimated from the concentration. Is it dry deposition or wet deposition or sum of the both? What was the deposition velocity and scavenging ratio? How much area was considered to estimate concentration into areal fluxes? All factors must be elaborated.

6. Are the measurements of atmospheric deposition, river inputs, N2 fixation, denitrification done simultaneously? If not, then how can one do mass balancing?

7. Models (mathematics) are useful to understand processes but one cannot ignore experimental results just because there are methodological issues. Because of methodological issues, experimentalists provide errors associated with estimates. Models also need to be verified with observations. Therefore, the criticism of experiments presented on page number 14 (first paragraph) is a bit overdone.

Minor comments:

Page 1, line 6-7: It is not just the nitrogen fixation and denitrification, riverine inputs and atmospheric deposition are equally important source of bioavailable nitrogen in coastal ecosystems.

Page 1, line 16 and wherever standard deviation is reported: standard deviation (error) should have only one significant digit and the mean value should be adjusted accordingly. See (Bevington & Robinson, 2003) for more details.

Page 1, line 19-20: Is 36% comparable with literature estimates based on direct measurements.

Page 2, line 5-8: "As these............…...eutrophic". This is intuitively correct but factually incorrect. We know that most of eutrophic oceans (for example the Arabian Sea and the eastern South Pacific) have the most intense denitrified zones, while the oligotrophic oceans (such as the North Atlantic) witness nitrogen fixation process.

Page 2, line 9-19: (Jickells et al., 2017) is an updated reference on this topic.

Page 2, line 21-24:, "This is particularly............…..........". Recent modelling and experimental studies, both have shown that the bioavailable nitrogen does not inhibit nitrogen fixation (Landolfi et al., 2015; Meyer et al., 2016)

Page 2, line 29: (Montoya et al., 1996) is an appropriate and original reference.

Page 3, line 1-6: Since the authors are listing issues with the N2 fixation estimation technique, they must also mention the bubble problem (Großkopf et al., 2012; White, 2012) and contamination problem (Dabundo et al., 2014).

Page 3, line 22: "is" should be replaced by "are"

Page 3, line 29: "However, such a model represents a simplified version. . . . . . . . . . . . . …..". Authors may refer to (Ramesh & Singh, 2010) for a complex isotope model based on Rayleigh isotope fractionation.

Page 4, line 19: sediment has appeared for the first time after its appearance in the abstract. Authors should make it clear in the first lines on the introduction that this study is based on the sediments.

Page 5, line 4: was should be replaced by were.

Page 6, line 11: Does t stand for ton in t N yr-1. State it accordingly.

Page 7, equation (6), Rivers also transport DON. This term must be included.

Page 9, line 8: How was the error estimated?

Page 11, line 4: Is 36% significantly different from 35%?

Table 1, column 3. Is it fractionation factor or just the delta value?

Table 2, How many data were used in this regression?

Fig. 3: What statistical analysis was done to test the difference (to obtain p value) between estimated and actual value?

References

Bevington, P. R., & Robinson, D. K. (2003). Data reduction and error analysis. McGraw-Hill.

Dabundo, R., Lehmann, M. F., Treibergs, L., Tobias, C. R., Altabet, M. A., Moisander, P. H., & Granger, J. (2014). The contamination of commercial 15N2 gas stocks with 15N–labeled nitrate and ammonium and consequences for nitrogen fixation measurements. PloS One, 9(10), e110335.

Großkopf, T., Mohr, W., Baustian, T., Schunck, H., Gill, D., Kuypers, M. M., . . . LaRoche, J. (2012). Doubling of marine dinitrogen-fixation rates based on direct measurements. Nature, 488(7411), 361–364.

Jickells, T., Buitenhuis, E., Altieri, K., Baker, A., Capone, D., Duce, R., . . . Zamora, L. M. (2017). A reevaluation of the magnitude and impacts of anthropogenic atmospheric nitrogen inputs on the ocean. Global Biogeochemical Cycles, 31(2), 289–305.

Landolfi, A., Koeve, W., Dietze, H., Kähler, P., & Oschlies, A. (2015). A new perspective on environmental controls of marine nitrogen fixation. Geophysical Research Letters.

Meyer, J., Löscher, C., Neulinger, S., Reichel, A., Loginova, A., Borchard, C., . . . Riebesell, U. (2016). Changing nutrient stoichiometry affects phytoplankton production, DOP accumulation and dinitrogen fixation–a mesocosm experiment in the eastern tropical North Atlantic. Biogeosciences, 13(3), 781–794.

Montoya, J. P., Voss, M., KÓŞhler, P., & Capone, D. G. (1996). A Simple, High-Precision, High-Sensitivity Tracer Assay for N2 Fixation. Applied and Environmental Microbiology, 62(3), 986–993.

Ramesh, R., & Singh, A. (2010). Isotopic fractionation in open systems: application to organic matter decomposition in ocean and land. Current Science, 98(3), 406–411.

Thamdrup, B., & Dalsgaard, T. (2002). Production of N2 through anaerobic ammonium oxidation coupled to nitrate reduction in marine sediments. Applied and Environmental Microbiology, 68(3), 1312–1318.

White, A. E. (2012). Oceanography: The trouble with the bubble. Nature, 488(7411), 290–291.

Please also note the supplement to this comment:
https://www.biogeosciences-discuss.net/bg-2017-418/bg-2017-418-RC1-supplement.pdf

---

## Author Comment (AC1) · 25 Jan 2018

We thank the reviewer for the constructive comments.

Major comments:

1. This model (equations presented on page 5-6) is very trivial. Nitrogen cycle is very dynamic, many components of the cycle are missing in the model, such as anaerobic ammonium oxidation that occurs in sediments (Thamdrup & Dalsgaard, 2002) and flux of the dissolved organic nitrogen (Jickells et al., 2017) to name a few. Moreover,

isotopic fractionation factors ($\varepsilon$) for these processes are different – not much is known about their precise values. None of the equations presented here have considered fractionation factors barring equation (3), and that too has inappropriately incorporated fractional factors. In addition, each term (expression) used in equations must be defined clearly, for example, what does mSources in equation (1) stand for? In a nutshell, equations need elaboration and model requires robustness.

*We agree that there are certainly other processes that contribute to the marine nitrogen cycle, and will include a more comprehensive discussion of these processes and the potential isotope effects associated. We note however that the net fractionation associated with these processes in sediment is small. Furthermore, all the processes discussed will result in an enrichment of the nitrogen isotope pool, and the most likely cause of the isotopically light signature of the nitrogen pool is due to nitrogen fixation, and this point will be reinforced.

2. These (model) equations are equations are time independent. We understand that biogeochemical processes are time dependent – for example N2 fixation is more in some season, while denitrification would dominate in some other season. So how good these isotope mass balance equation can represent such processes?

*It is true that biogeochemical processes are time dependent, however, using instantaneous rate measures to describe the cycling of nitrogen over the annual cycle is problematic i.e. how well will the data on any given day represent actual processing rates. The advantage of using stable isotopes in such a study is that they represent integrated measures of the accumulated pool and hence provide an insight into the longer-term behaviour of nitrogen cycling and the implications over the course of a year, which was the intention of this manuscript. Additional text will be added to emphasize this point

3. It is stated that equations (2) and (3) provide equation (4) (Page 5, line 30) but I guess equations (1) and (3) provide (4).

*Eq (4) is created by substituting Eqs. (2) and (3) into Eq. (1). This will be re-worded

in the revised manuscript.

4. Section 2.2.1 on atmospheric deposition. Dissolved organic nitrogen in an important source of nitrogen that has been ignored.

*Dissolved organic nitrogen can comprise up to 50% of total atmospheric deposition in this region (Lansdown, 2009). Because of the unknown bioavailability of this fraction and also the fact that atmospheric deposition itself only contributes <5% to the total nitrogen input, we believe this will make little difference to the budget. This assumption will be stated in the revised version.

5. In the same section, how was atmospheric flux estimated from the concentration. Is it dry deposition or wet deposition or sum of the both? What was the deposition velocity and scavenging ratio? How much area was considered to estimate concentration into areal fluxes? All factors must be elaborated.

*This was estimated as bulk atmospheric deposition. We will provide a more comprehensive discussion of how the atmospheric flux was estimated in the methods section.

6. Are the measurements of atmospheric deposition, river inputs, N2 fixation, denitrification done simultaneously? If not, then how can one do mass balancing?

*Yes, with the exception of the concentrations that were used in calculating the input of DIN from atmospheric deposition (these were based on separate studies), all other measurements were undertaken concurrently. We will make further clarifying remarks explicitly dealing with this point in the methods section

7. Models (mathematics) are useful to understand processes but one cannot ignore experimental results just because there are methodological issues. Because of methodological issues, experimentalists provide errors associated with estimates. Models also need to be verified with observations. Therefore, the criticism of experiments presented on page number 14 (first paragraph) is a bit overdone.

*We agree this criticism is over emphasized and we will tone this back. We will emphasize this model matches closely with our previous experimental results (Russell et al., 2016) and also highlight how this approach integrates over time (and therefore time independence) of the mass-balance model.

Minor Comments:

All the minor comments will be addressed in the revised manuscript.

References Cited:

Lansdown, K. P.-M.: Biogeochemistry of nitrate in headwater streams and atmospheric deposition of the Dandenong Ranges, PhD thesis, Monash University, 2009.

Russell, D. G., Warry, F. Y., and Cook, P. L. M.: The balance between nitrogen fixation and denitrification on vegetated and non-vegetated intertidal sediments, Limnology and Oceanography, 61, 2058-2075, doi: 10.1002/lno.10353, 2016.

---

## Referee Comment (RC2) · Anonymous Referee #2 · 19 Mar 2018

This manuscript estimated the N supply by nitrogen fixation under an assumption of balanced nitrogen influx (source) and outflux (sink) in the intertidal embayment, and addressed its validity by comparing 15N of sediments between the model derived and measured values. It's interesting approach, but I have serious concerns about the steady state in the study area, and the fluxes and isotopes used in the model (see the following comments). While I appreciate the effort of the work presented, I cannot recommend this manuscript for publication unless these concerns are solved clearly.

Major comments: 1. I doubt the steady state within the small and complicated bay

that authors assumed. This requires no change of nitrogen budget (pool size) in the system. Authors should discuss about major nitrogen pools, their sizes and temporal change of them.

2. Authors also assumed the pseudo closed system in which all nitrogen supplied is consumed by the sink processes (sediment burial, denitrification, algal assimilation) within the study area. This implied that any riverine nitrate is not transported out of the bay. Although authors pointed extremely low concentration of dissolved inorganic nitrogen in the bay, Russell et al. (2016) reported the range of 0.2-5 uM of nitrate. The spatial distribution of nitrate should be presented along the rivers-the inside bay-out of the bay, then mixing of freshwater and seawater to be discussed. Authors also suggested no flux between inside and out of the bay for the particulate nitrogen because of similar concentration and its isotope between them. However, many scientists (e.g. Sukigara & Saino, 2006, Geophysical Research Letters, 33, L09607) have stressed a significant transport of resuspended sediments as nitrogen flux from the bay to open ocean. This possibility should be examined in this study. If any riverine materials including nitrogen is not transported, dissolved inorganic nitrogen out of the bay should originate from open ocean. Is it true?

3. Authors assumed that almost riverine nitrate are assimilated by phytoplankton and seagrass. Ultimately, I think, these organic nitrogen is decomposed into inorganic nitrogen (ammonium and nitrate). A part of them can be buried into the bottom. The nitrate regenerated from algal organic nitrogen can be consumed by denitrification. These processes links each other complicatedly. It's impossible to estimate their independent fluxes, especially in annual scale.

4. Authors seem to confound the 15N of removed nitrogen with the isotopic fractionation associated with the removal (sink) processes. The isotopic fractionation ($\varepsilon$) is expressed as 15N difference between substrate and product of the process. Therefore, equation (3) was inadequate. Furthermore, I have some concerns about 15N of removed nitrogen used in the model. As for denitrification, 15N of 3.5 ‰ is used by

referring Brandes & Devol (2002) in which they assumed 15N of typical oceanic nitrate with 5 ‰ and $\varepsilon$ with 1.5 ‰. Meanwhile, authors suggested that riverine nitrate 15N was 12.6 ‰ (P7_L24), which looks to conflict with nitrate 15N of 5 ‰. Authors should explain the origin of nitrate in the bay. As for algal assimilation, it's okay with $\varepsilon$ of 4 ‰. The 15N of assimilated nitrogen, however, to be calculated from nitrate 15N minus $\varepsilon$. If assuming riverine nitrate with 12.6 ‰ as a major substrate, it corresponds with 8.6 ‰ (= 12.6 - 4.0). This would lower the sediment 15N estimated from your model.

5. I'm afraid I can't understand the model calculation in this study. The sediment 15N derived from this model were shown in Fig. 3 and S1, which are the output of 10,000 iterations (P10_L16). I suspect that these outputs are same with the result of sensitivity analysis, illustrated in Fig. S2 and S3. If so, I cannot find any significance of the average and the standard deviation of this result because, I think, they do not support the validity of model output.

---

## Author Comment (AC2) · 7 Apr 2018

*We thank the reviewer for their comments. In reading these comments, we realise we have not articulated the model or the key assumptions behind it clearly enough. In the revised version of the paper we will stress that the model is essentially an isotope mass balance model used routinely in ecological studies. The relative contribution to each end-member is estimated by comparing mass weighted isotope sources (the rivers, nitrogen fixation, the ocean and the atmosphere). The key assumptions are as follows:

[Figure]

1. All sources of nitrogen are assimilated into the solid pool which is well mixed and then some loss though denitrification. This is justified by the fact that it is a macrotidal system and that it has a relatively uniform marine salinity. 2. Algae/vegetation assimilates all dissolved nitrogen hence there is no fractionation associated with this process, and eventually all the nitrogen is returned to the sedimentary nitrogen pool. 3. We note that steady state conditions are not required for this model. A mixing model such as ours does not require steady state, but does assume the nitrogen pool is well mixed in this system, and therefore the isotopic values will not be affected if there is export of material from the bay. Our measurements of the outgoing N flux support this. For more information on mixing models such as this, please see Fry (2006).

This manuscript estimated the N supply by nitrogen fixation under an assumption of balanced nitrogen influx (source) and outflux (sink) in the intertidal embayment, and addressed its validity by comparing 15N of sediments between the model derived and measured values. It's interesting approach, but I have serious concerns about the steady state in the study area, and the fluxes and isotopes used in the model (see the following comments). While I appreciate the effort of the work presented, I cannot recommend this manuscript for publication unless these concerns are solved clearly. Major comments:

1. I doubt the steady state within the small and complicated bay that authors assumed. This requires no change of nitrogen budget (pool size) in the system. Authors should discuss about major nitrogen pools, their sizes and temporal change of them.

*The key assumption in this study is that the nitrogen pool in the bay is well mixed. Consequently, we do not need to assume that steady state conditions exist within the bay, this is because once the isotopes are well mixed, and this will not affect isotope values in the case of export of material from the bay. The revised manuscript will clearly articulate this point and provide evidence in the support of this assumption. This includes: 1. The observation that the d15N of sediment within the bay was relatively uniform. 2. The macrotidal and uniformly marine nature of the bay.

[Figure]

2. Authors also assumed the pseudo closed system in which all nitrogen supplied is consumed by the sink processes (sediment burial, denitrification, algal assimilation) within the study area. This implied that any riverine nitrate is not transported out of the bay. Although authors pointed extremely low concentration of dissolved inorganic nitrogen in the bay, Russell et al. (2016) reported the range of 0.2-5 uM of nitrate. The spatial distribution of nitrate should be presented along the rivers-the inside bay-out of the bay, then mixing of freshwater and seawater to be discussed.

*Samples collected at the ingoing and outgoing tide suggest no significant DIN import or export from the bay. Therefore, we do not believe that discussion of nitrate gradients from rivers to bay to out-of-bay are relevant. The revised manuscript will illustrate this point more clearly.

Authors also suggested no flux between inside and out of the bay for the particulate nitrogen because of similar concentration and its isotope between them. However, many scientists (e.g. Sukigara & Saino, 2006, Geophysical Research Letters, 33, L09607) have stressed a significant transport of resuspended sediments as nitrogen flux from the bay to open ocean. This possibility should be examined in this study. If any riverine materials including nitrogen is not transported, dissolved inorganic nitrogen out of the bay should originate from open ocean. Is it true?

*Whilst the above scenario is possible, our measurements of particulate N as described by the reviewer suggest that this is not the case for our system. Instead, we suggest that exported particulate nitrogen originates from sources such as remineralised and nitrified ammonium from the buried/sediment pools. As a consequence, it will have been well mixed into the bay isotope pool, and therefore even if there is net import/export of material this will not affect the outcome of our model. Nonetheless, the reviewers point is a valid one, and we will include discussion of resuspension, and why it does not need to factor in our model, in the revised manuscript.

3. Authors assumed that almost riverine nitrate are assimilated by phytoplankton and

seagrass. Ultimately, I think, these organic nitrogen is decomposed into inorganic nitrogen (ammonium and nitrate). A part of them can be buried into the bottom. The nitrate regenerated from algal organic nitrogen can be consumed by denitrification. These processes links each other complicatedly. It's impossible to estimate their independent fluxes, especially in annual scale.

*We assumed that the riverine nitrogen was either assimilated, then mixed into the sediment pool, or that refractory particulate nitrogen was mixed into the sediment pool, contributing to the isotope signature in that manner. There are many cycling processes – including decomposition to organic N as the reviewer suggests, but so long as the assimilated N forms a well-mixed pool, these processes do not fractionate much and hence do not affect the d15N of the isotope pool.

4. Authors seem to confound the 15N of removed nitrogen with the isotopic fractionation associated with the removal (sink) processes. The isotopic fractionation ($\varepsilon$) is expressed as 15N difference between substrate and product of the process. Therefore, equation (3) was inadequate.

*We thank the reviewer for pointing this out, equation 3 will be re-written to accurately reflect the removal of nitrogen due to denitrification. We also note that there is an error in Figure S3 regarding the effect of the denitrification fractionation factor on the d15N of the sediment pool (the axes are reversed). This mistake will also be rectified in the revised manuscript.

Furthermore, I have some concerns about 15N of removed nitrogen used in the model. As for denitrification, 15N of 3.5 ‰ is used by referring Brandes & Devol (2002) in which they assumed 15N of typical oceanic nitrate with 5‰ and $\varepsilon$ with 1.5‰ Meanwhile, authors suggested that riverine nitrate 15N was 12.6‰ (P7_L24), which looks to conflict with nitrate 15N of 5‰ Authors should explain the origin of nitrate in the bay. As for algal assimilation, it's okay with $\varepsilon$ of 4‰ The15N of assimilated nitrogen, however, to be calculated from nitrate 15N minus $\varepsilon$. If assuming riverine nitrate with 12.6 ‰ as a major

substrate, it corresponds with 8.6 ‰ (= 12.6 - 4.0). This would lower the sediment 15N estimated from your model.

*We thank the reviewer for raising this point, however on reflection, the effect of algal assimilation is likely to be insignificant. This is because there is little evidence of riverine-derived nitrogen being exported from the bay (see above), which suggests that it is either being directly bound up in this sedimentary pool, or there is algal assimilation. If algal assimilation is occurring, this will have limited effects as ultimately this nitrogen will re-enter the sediment pool as algae die and are buried. A similar argument also exists for vegetative assimilation. In our revised manuscript, this point along with all relevant assumptions will be more clearly articulated and the proposed isotope balance modified accordingly.

5. I'm afraid I can't understand the model calculation in this study. The sediment 15N derived from this model were shown in Fig. 3 and S1, which are the output of 10,000 iterations (P10_L16). I suspect that these outputs are same with the result of sensitivity analysis, illustrated in Fig. S2 and S3. If so, I cannot find any significance of the average and the standard deviation of this result because, I think, they do not support the validity of model output.

*We agree with the reviewer the overarching rationale behind what was attempted here has not be sufficiently explained, and as a consequence the approach used and its meaning are confusing. At its core, this study was using an isotope mixing model to estimate the nitrogen isotope value of the sedimentary nitrogen pool based upon prior measurements of nitrogen processing rates in the bay. A comparison between these estimated isotope values and values that were taken independently will be undertaken to assess how good the previous rate measurements were (particularly nitrogen fixation). This in turn will also allow for a commentary surrounding the importance of nitrogen fixation in coastal embayments to be undertaken. The revised version of this manuscript will ensure that these points are clearly addressed.

References Cited:

Fry, B.: Stable Isotope Ecology, Springer, New York, 2006.